# Water-stable porous Al$_{24}$ Archimedean solids for removal of trace iodine

Ya-Jie Liu[1,2], Yi-Fan Sun[1], Si-Hao Shen[1], San-Tai Wang[1], Zhuang-Hua Liu[1], Wei-Hui Fang [1]✉, Dominic S. Wright [3] & Jian Zhang [1]✉

In this paper, we report a unique type of core-shell crystalline material that combines an inorganic zeolitic cage structure with a macrocyclic host arrangement and that can remove trace levels of iodine from water effectively. These unique assemblies are made up of an inorganic Archimedean truncatedhexahedron (*tcu*) polyhedron in the kernel which possesses six calixarene-like shell cavities. The cages have good adaptability to guests and can be assembled into a series of supramolecular structures in the crystalline state with different lattice pore shapes. Due to the unique core-shell porous structures, the compounds are not only stable in organic solvents but also in water. The characteristics of the cages enable rapid iodine capture from low concentration aqueous I$_2$/KI solutions (down to 4 ppm concentration). We have studied the detailed process and mechanism of iodine capture and aggregation at the molecular level. The facile synthesis, considerable adsorption capacity, recyclability, and β- and γ-radiation resistance of the cages should make these materials suitable for the extraction of iodine from aqueous effluent streams (most obviously, radioactive iodide produced by atomic power generation).

Volatile radioactive species present in water cooling streams from nuclear fission reactors pose a serious threat to human health and the environment. Radioactive $^{131}$I ($t_{1/2}$ = 8.02 days) and $^{129}$I ($t_{1/2}$ = 15.7 million years) which are common decay products both pose a significant long-term health risk due to β and γ radioactive decay[1–3]. Therefore, there is an urgent need for materials that can capture iodine from aqueous effluents. Recent advances in this field together with synthetic chemistry have led to the development of materials capable of the removal of radioactive iodine. Currently, solid-phase adsorption includes ion exchange[4], forming precipitates[5], and chemical bonding[6], which have notable advantages due to easy handling, avoidance of secondary pollution, and high removal efficiency[7]. Solid crystalline materials with long-range order are useful models for elucidation of the mechanism of radioactive element capture at the molecular level and may help in the design and synthesis of advanced materials[8–10]. Some crystalline materials, such as zeolites[11,12], metal-organic frameworks (MOFs)[13,14],

and hydrogen-bonded organic frameworks (H$_C$OFs)[15], have been applied to water-phase iodide ion capture.

Supramolecular materials and cage compounds have also illustrated the potential for iodine removal as a result of their intramolecular and intermolecular host cavities for guest inclusion[16–20]. Considering ion exchange, cationic aluminum oxo clusters are more suitable candidates for capturing iodide ions than extensively studied polyoxometalate anions[21,22]. However, despite their wide application in water treatment (e.g., toxic arsenate adsorption), there are no studies of iodide anion removal from water reported. This may be due to their dense Keggin-type and Brucite-like cage[23–25] structures, the lack of suitable sites (such as conjugated groups) and appropriate cavities for absorption, and their poor crystallinity (i.e., limiting studies of the binding mechanisms from crystallographic analysis). As noted above, well-studied zeolites are stable with high adsorption and separation properties, while supramolecular cage materials have confined cavities

[1]State Key Laboratory of Structural Chemistry, Fujian Institute of Research on the Structure of Matter, Chinese Academy of Sciences, Fuzhou, Fujian 350002, P. R. China. [2]University of Chinese Academy of Sciences, Chinese Academy of Sciences, Beijing 100049, P. R. China. [3]The Yusuf Hamied Chemistry Department, Cambridge University, Lensfield Road, Cambridge CB2 1EW, UK. ✉e-mail: fwh@fjirsm.ac.cn; zhj@fjirsm.ac.cn

and abundant capture sites. If the characteristics of the two can be combined (microporous zeolite channels and supramolecular macrocyclic sites), it might be possible to form a unique type of porous material for the efficient removal of iodide ions in water.

Based on the above considerations and our previous work on aluminum molecular rings[26–29], we herein report the aggregation of the aluminum molecular rings into cationic core–shell $Al_{24}$ Archimedean solids and their performance in removing iodide from water. The $Al_{24}$ Archimedean solids consist of a pure-inorganic truncatedhexahedron (*tcu*) cage in the core together with six calixarene-like macrocyclic cavities. Notably, this assembly is an unprecedented combination of an inorganic zeolite-like cage with macrocyclic units. The macrocyclic cavities are situated on six faces of the *tcu* polyhedron, consequently, we refer to this unique class of compounds as "aluminum macrocycle-faced cages" (abbreviated as **AlMCs**). The **AlMCs** are well adapted to the guests, exhibiting key features required for iodide extraction—self-assembly, recrystallizability, and reversible ion exchange (Fig. 1). They exhibit high chemical stability, including excellent water stability, and can be prepared in large quantities. The $Al_{24}$ cation units present not only exhibit a fast adsorption response but also a low capture concentration for iodine ions, indicating their potential as iodide absorbents in water. The iodide capture and aggregation processes have been revealed at the molecular level using crystal-to-crystal diffraction studies.

## Results

### Archimedean $Al_{24}$ cage structure

All of the **AlMC** host–guest complexes have similar molecular units in which the host is the cation $[Al_{24}(BA)_{12}(EtO)_{24}(OH)_{32}]^{4+}$ ($Al_{24}$) (BA = benzoate), constructed from truncated hexahedron (*tcu*) made up of six octagonal $Al_8$ faces and eight triangular $Al_3$ faces (Fig. 2a, Supplementary Movie 1). Of all the 13 Archimedean polyhedra the *tcu* is the only one composed of triangles and octagons (Supplementary Figs. 1–4). In previous rarely reported *tcu* cages, organic ligands are used as edges, such as in $[V_{24}O_{24}(C_4O_4)_{12}(OCH_3)_{32}]^{8-}$ reported by Hartl et al.[30], and the $Ag_{24}L_{16}$ cage reported by Fujita et al.[31]. Notably, $Al_{24}$ is a rare example possessing a purely inorganic *tcu* cage. There are four hydroxyl groups that point towards the center of each octagonal face of the $Al_{24}$ unit (Supplementary Fig. 5), and one hydroxyl group on every triangular face (Supplementary Fig. 6), forming an internal inorganic cubic cavity (the available volume being ~320 $\AA^3$) (Supplementary Fig. 7). The $Al_{24}$ core is capped by six calixarene-like macrocyclic fragments over each of the octagonal faces of the *tcu* core. These macrocyclic units are composed of 8 $Al^{3+}$ cations which are bonded together using a combination of $\mu_2$-OH, benzoate, and ethoxide groups. The dimensions of the macrocyclic apertures are very similar to that found in calix[4]arene (the height, lower diameter, and upper diameter are, respectively, $4.83 \times 4.97 \times 19.18 \, \AA^3$ vs. $4.51 \times 4.48 \times 16.16 \, \AA^3$ for calix[4]arene) (Supplementary Fig. 8). Hence, the whole $Al_{24}$ arrangement has a molecular diameter of ~ 2.0 nm (Supplementary Figs. 9–11). Compared with traditional metal-organic cages with a single cavity[32–34], the $Al_{24}$ cage, therefore, has a 'two-tier' cavity arrangement, which combines the characteristics of an inorganic metal cage with that of a calixarene, and possesses inorganic hydrophilic and organic hydrophobic cavities (Supplementary Movie 2).

As shown in Fig. 2b, the molecular polyhedron possesses a highly symmetrical geometry and a five-level nested structure (Supplementary Fig. 12), i.e., from the inner inorganic component to the outer organic components, $O_8$ *cube* @ *tro* @ *tcu* @ *rco* @ *cuo*. For the inner inorganic fragment, eight $\mu_3$-OH groups form a centered $O_8$ *cube* (with a diameter of 4.47 Å). The second $O_{24}$ shell is a truncatedoctahedral (*tro*) shell made up of 24 $\mu_2$-OH groups (with a diameter of 8.16 Å). As far as the organic ligands are concerned, 24 ethoxide groups are situated on the edges of triangular faces constituting an $(OR)_{24}$ rhombicuboctahedron (*rco*) subunit (with a diameter of 9.25 Å), while the benzoate ligands bridge adjacent $Al_3(\mu_3$-OH) segments and assemble into a $(BA)_{12}$ cuboctahedron (*cuo*) arrangement (with a diameter of 19.18 Å).

The peripheral macrocyclic units of the $Al_{24}$ host can trap a large array of guests of different sizes, shapes, and charges, including neutral water molecules, ethanol, n-propyl alcohol, nitrate, and halide anions (Fig. 3a, Supplementary Figs. 13–19). These guest molecules form H-bonding interactions with the four OH groups of the macrocyclic units with O–H···O range 2.823–3.354 Å and O–H···X (X = Cl, Br, I) range 3.131–3.728 Å (Supplementary Figs. 20–25), which are close to those reported in the literature[35–38]. The six guests that are accommodated form an octahedron (sizes: 11.19 Å × 7.92 Å–13.76 Å × 9.73 Å, Supplementary Fig. 13). The depth and aperture size of these $Al_8$ macrocyclic subunits vary depending on the guest present (depth: 4.58–5.49 Å, aperture: 16.42–18.76 Å) (Supplementary Table 1).

The interaction of the $Al_{24}$ cations with these guests generates a diverse range of supramolecular lattice arrangements, including monoclinic **AlMC-1** (space group, C2/c) and **AlMC-3** (space group, P2₁/n), triclinic **AlMC-2** and **AlMC-4** (space group, P−1), trigonal **AlMC-7** (space group, R−3), and cubic **AlMC-5** (space group, Ia-3), as well as **AlMC-6** (space group, Im-3m) (Fig. 3b). Their supramolecular packing in the solid state correlates with the crystal morphology (Supplementary Fig. 26). Typically, the solid-state packing involves the back-to-back alignment of two $Al_8$ macrocyclic subunits on adjacent $Al_{24}$

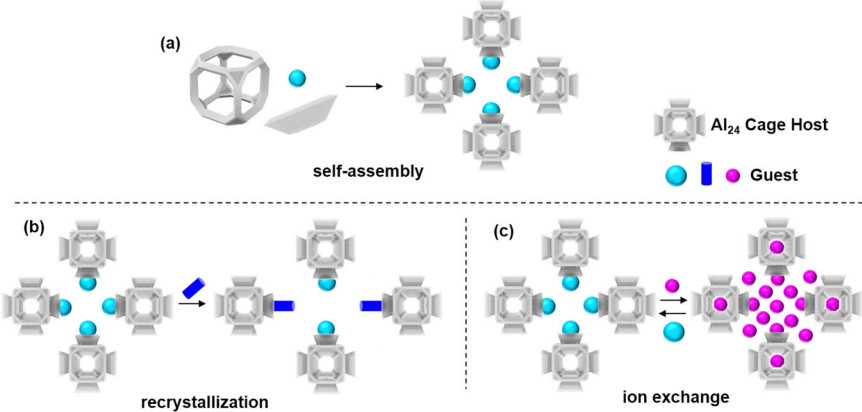

**Fig. 1 | The supramolecular assembly of the $Al_{24}$ cages. a** One-step **AlMC** self-assembly results in the integration of an inorganic *tcu* cage containing calixarene-like macrocyclic units. **b** Recrystallization of the **AlMC** results in a lattice with embedded guest molecules. **c** Reversible single crystal-to-single crystal ion exchange process of the **AlMC**.

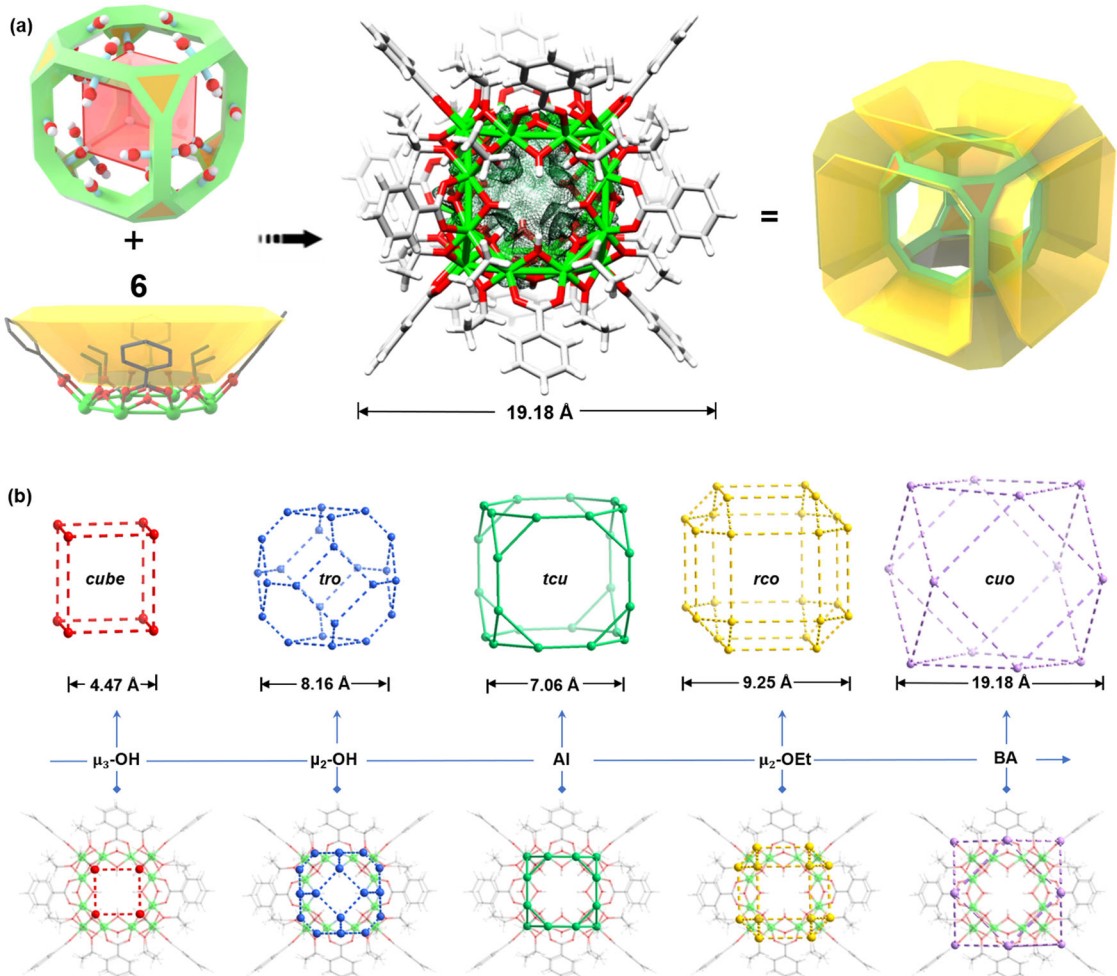

**Fig. 2 | The molecular structure of the Al$_{24}$ Archimedean cage. a** Illustration of the assembly and structural model of the inner inorganic *tcu* core and calixarene-like organic subunits (atom color code: Al: bright green; O: red; C: gray; H: white). The size of the *tcu* cage is ca. 1.9 nm based on the distance between the BA ligands. **b** Quintuple structural symmetric anatomy of the Al$_{24}$ Archimedean solid. The inner shell is an O$_8$ *cube*, the second shell is an O$_{24}$ truncatedoctahedral (*tro*) cage, the third shell is an Al$_{24}$ truncatedhexahedron (*tcu*) cage, the fourth shell is an (OR)$_{24}$ rhombicuboctahedron (*rco*) cage, and the outermost shell is a (BA)$_{12}$ cuboctahedron (*cuo*) cage.

cages which accommodate a pair of guests (Supplementary Figs. 27–32). The shape and size of this coordination pocket and the alignment of the macrocyclic units are highly dependent on the guests (Supplementary Figs. 33–38), being spindle- (Supplementary Fig. 33a), peanut- (Supplementary Fig. 33b), or Z-shaped (Supplementary Fig. 33c), and with a distance between the two macrocyclic units varying in the order H$_2$O cavity (4.794 Å) < Cl$^-$ cavity (7.625 Å) < ethanol cavity (7.625–7.706 Å) < n-propanol cavity (7.892 Å) < Br$^-$ cavity (9.156 Å) < NO$_3^-$ cavity (10.483–10.883 Å) (Supplementary Figs. 27–32). The presence of small anions as guests (Cl$^-$, Br$^-$ and I$^-$) not only influences the solid-state packing of the cages but these anions are also present within the center of the cage (H-bonded within the eight OH groups of the central cubic O$_8$ unit). **AlMC-1** to **AlMC-5** contain disordered NO$_3^-$ anions within their O$_8$ cavities, with the number decreasing from 2 (in **AlMC-1** to **AlMC-3**) to 1 (in **AlMC-4** and **AlMC-5**) in the presence of halide ions (Supplementary Figs. 39–42). In **AlMC-6** containing no NO$_3^-$, this site is occupied by an I$^-$ anion (Supplementary Fig. 42c). **AlMC-7** contains a cationic macrocycle in its supramolecular arrangement [(Al$_6$(BA)$_6$(OEt)$_6$(NO$_3$)$_2$)$_{0.5}$]$^{2+}$ which links units of Al$_{24}$ together (Supplementary Figs. 43–49). The Al$_6$-ring unit of the latter contains 6 Al$^{3+}$ centers held together by 6 benzoate ligands, 6 alk-oxides, and two NO$_3^-$ anions. To the best of our knowledge, such a cationic {Al$_6$} ring has not been reported previously, although similar

neutral Al-macrocycles have been observed by us[26,28]. The Al$_{24}$ cations are surrounded by six Al$_6$-rings in **AlMC-7** (Supplementary Figs. 13 and 46), thus, there are dual-Platonic octahedra present, one is formed by the six capping ethanol guests, and the other is created via six Al$_6$ rings (Supplementary Fig. 46). The isolation of **AlMC-7**, containing these Al$_6$ macrocyclic cations, provides some potential insight into the mechanism of formation of the Al$_{24}$ cation itself, which potentially results from the condensation of these smaller rings during the reaction.

Supramolecular assembly can also be achieved by recrystallization from acetonitrile. An interesting stack transformation occurs on crystallization of **AlMC-5** from *Ia*-3 (cube crystals) to *P*2$_1$/n in **AlMC-8** (parallelogram crystals) from acetonitrile (Fig. 3, Supplementary Figs. 50–53), with the guests experiencing a slight adjustment in coordination environment. The phase purity for **AlMC-1** to **AlMC-8** was validated by their powder X-ray diffraction (PXRD) patterns (Supplementary Figs. 54–61). The presence of some of the anion guests can be confirmed by energy-dispersive X-ray dispersive spectroscopies (EDS) (Supplementary Figs. 62–69) and Fourier transform infrared (FT-IR) spectroscopies (Supplementary Figs. 70–77). The apparent band gaps for colorless compounds (except **AlMC-6**) are in the range of 4.2–4.3 eV (Supplementary Figs. 78–85). Furthermore, thermogravimetric analysis (TGA) of **AlMC-1** to **AlMC-7** showed that these cages

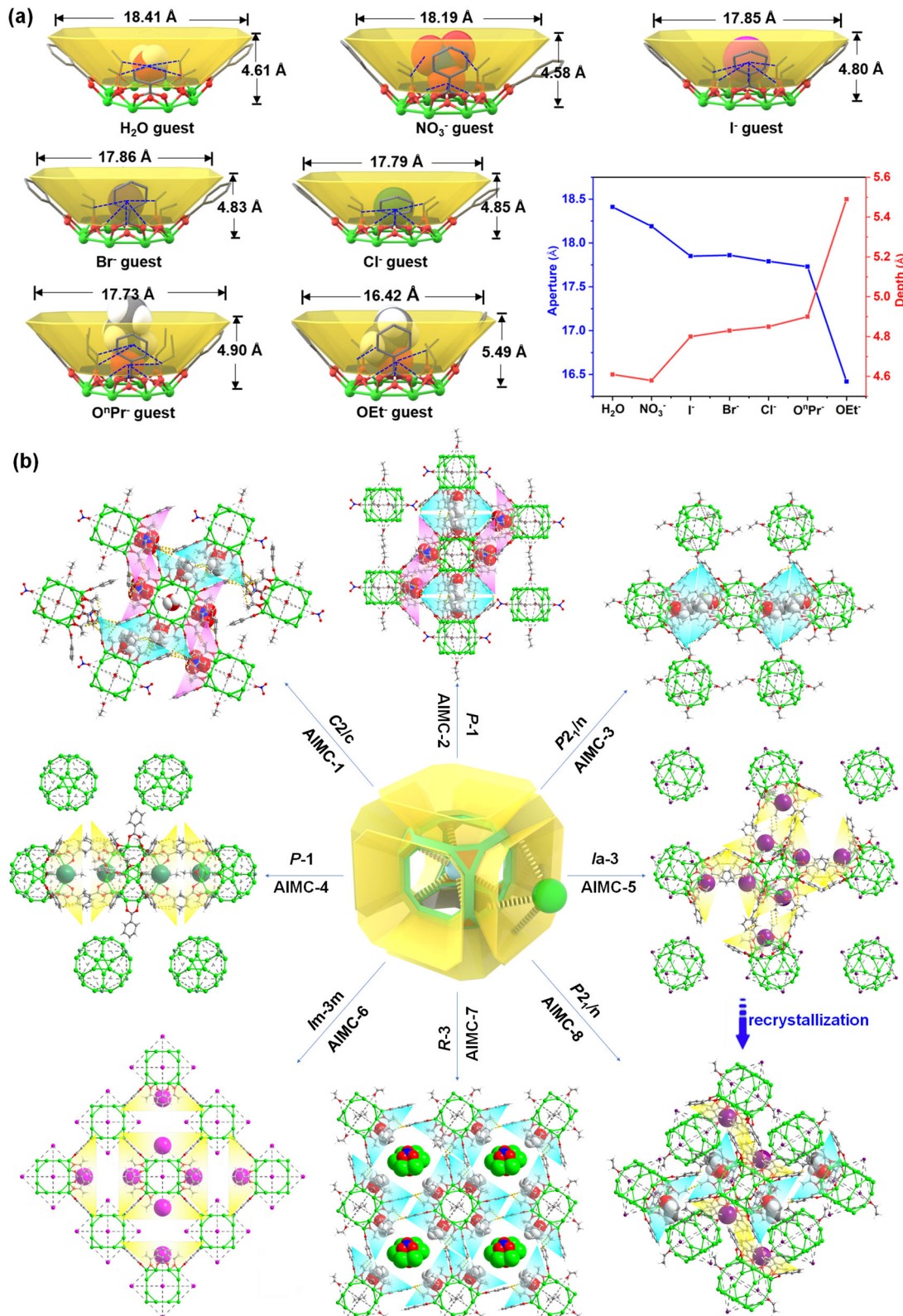

**Fig. 3 | Supramolecular lattice assemblies between the cationic $Al_{24}$ host and a variety of guests. a** The $Al_8$ macrocyclic subunits adapt to the accommodated $H_2O$ (in **AlMC-1**), $NO_3^-$ (in **AlMC-1**), $I^-$ (in **AlMC-6**), $Br^-$ (in **AlMC-5**), $Cl^-$ (in **AlMC-4**), $O^nPr^-$ (in **AlMC-2**) and $OEt^-$ (in **AlMC-3**) guests. The blue dotted lines indicate that there are strong hydrogen bond interactions between two atoms (the details of the hydrogen bond interactions are provided in Supplementary Figs. 20–25). **b** Packing diagrams of **AlMC-1**−**AlMC-8**. Hydrogen-bond interactions between neighboring $Al_{24}$ units are shown with yellow dotted lines. ($NO_3^-$ macrocyclic cavities: pink; alcohol/alkoxide macrocyclic cavities: blue; halogen ion macrocyclic cavities: yellow). The molecular formulae of **AlMC-1**−**AlMC-8** are, respectively: $Al_{24} \cdot 4NO_3^- \cdot 2\text{-}HOEt \cdot 2H_2O$ (**AlMC-1**), $Al_{24} \cdot 4NO_3^- \cdot 4HO^nPr$ (**AlMC-2**), $Al_{24} \cdot 2NO_3^- \cdot 4HOEt \cdot 2OEt^-$ (**AlMC-3**), $Al_{24} \cdot NO_3^- \cdot 3Cl^-$ (**AlMC-4**), $Al_{24} \cdot NO_3^- \cdot 3Br^-$ (**AlMC-5**), $Al_{24} \cdot 4I^-$ (**AlMC-6**), $Al_{24} \cdot H\text{-}NO_3^- \cdot 6OEt^- \cdot (Al_6(BA)_6(OEt)_6(NO_3)_2)_{0.5}$ (**AlMC-7**), $Al_{24} \cdot NO_3^- \cdot 2Br^- \cdot OEt^-$ (**AlMC-8**) (Al: bright green; O: red; C: gray; H: white; N: blue; Cl: sea green; Br: purple; I: pink).

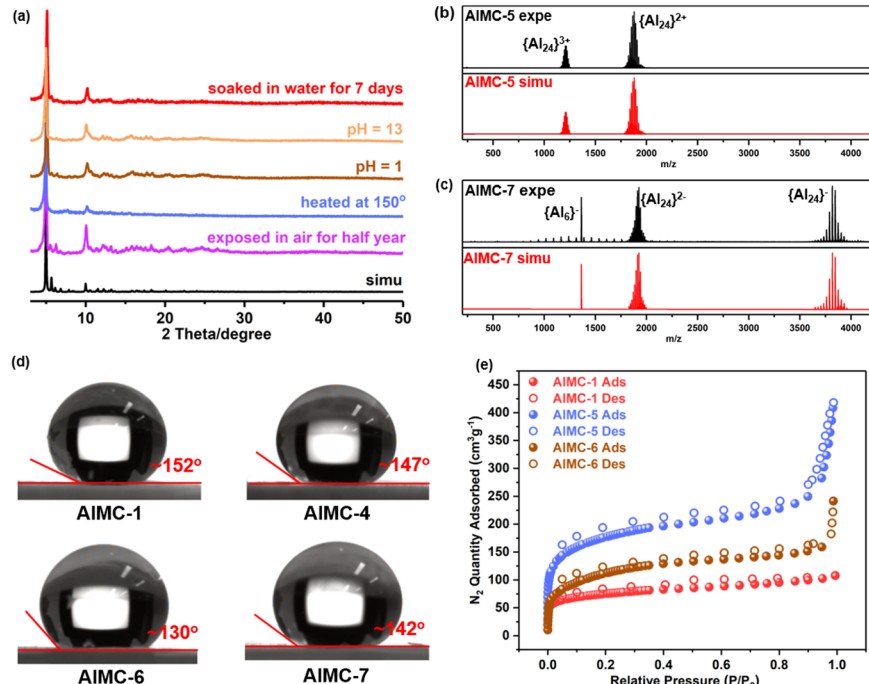

**Fig. 4 | Stability, wettability, and porosity measurements on selected cages.**
**a** Structural stability of **AlMC-1** under different conditions. **b** Solution stability of
**AlMC-5**: experimental and simulated mass spectra in MeCN under positive mode.
**c** Solution stability of **AlMC-7**: experimental and simulated mass spectra in a mixed
solvent of MeCN and DMF (volume ratio: 20:1) under negative mode. **d** The wett-
abilities of **AlMC-1**, **AlMC-4**, **AlMC-6**, and **AlMC-7**. **e** The $N_2$ gas sorption isotherms
at 77 K for **AlMC-1**, **AlMC-5**, and **AlMC-6**. Source data are provided as a Source
Data file.

remain stable up to 170–180 °C under $N_2$ atmosphere (Supplementary
Figs. 86–92).

## Stability characteristics

Stability is a critical issue that needs to be considered in systems with
real-world applications. All of the compounds exhibit high air stability
(e.g., **AlMC-1** is stable for more than half a year, Fig. 4a), thermal sta-
bility (e.g., **AlMC-1** remains crystalline at 150 °C, Fig. 4a and Supple-
mentary Fig. 93), and a high degree of chemical stability. They are
stable in common low-polarity organic solvents (Supplementary
Figs. 94–100), and soluble in highly polar aprotic solvents, like acet-
onitrile ($CH_3CN$), dimethylformamide (DMF) and dimethylsulfoxide
(DMSO) (Supplementary Table 2). The stability of the cationic $Al_{24}$ unit
was monitored by ESI-MS spectra (Fig. 4b, c and Supplementary
Figs. 101–104, Supplementary Tables 3–8) and $^1H$ NMR spectroscopic
analysis (Supplementary Fig. 105) by dissolving single crystals in
acetonitrile or DMSO[39]. For example, ESI-MS analysis of **AlMC-5** gave a
spectrum with two sets of dominant peaks assigned to $[Al_{24}(BA)_{12}(\mu_3-OH)_8(\mu_2-OH)_x(OEt)_{48-x}\cdot NO_3]^{3+}$ ($x = 25$–36) and $[Al_{24}(BA)_{12}(\mu_3-OH)_8(\mu_2-OH)_x(OEt)_{48-x}\cdot 9CH_3CN\cdot NO_3\cdot Br]^{2+}$ ($x = 36$–48) due to the loss of ethanol
(Fig. 4b)[40]. The other major consecutive peaks in **AlMC-5** and their
formulae are provided in Supplementary Table 7. In addition, the
presence of the $[Al_6(BA)_6(NO_3)_2(OH)_{11}\cdot(CH_3CN)_4]^-$ ion observed in the
negative-ion ESI-MS of **AlMC-7** strongly supported the crystallographic
results (exp: 1363.10; cal: 1363.17) (Fig. 4c and Supplementary Table 8).

Water stability is an important property with respect to aqueous-
phase iodide capture (see later in this paper). Water stability was
evaluated by immersing as-prepared crystals of **AlMC-1** in water at
different pH values for 24 h and was confirmed by the unchanged
PXRD pattern (Fig. 4a). In order to evaluate the long-term stability,
crystals of **AlMC-1** were soaked in water for 7 days at room tempera-
ture. Combined PXRD (Fig. 4a), Fourier transform infrared spectro-
scopy (FTIR) (Supplementary Fig. 106), and the preservation of the
morphology of the transparent crystals (Supplementary Fig. 107)

indicated its good water stability. In addition, the nearly identical cell
parameters encourage us to collect the single-crystal X-ray diffraction
data on these water-exposed crystals (Supplementary Fig. 108). We
find that in the product (**AlMC-1a**) the complete $Al_{24}$ ***tcu*** skeleton is
preserved (Supplementary Fig. 109). The water stability of **AlMC-1** is
presumably partly related not only to the presence of hydrophobic
ligands which shield the internal $Al^{3+}$ ions from attack by $H_2O$ (Fig. 4d
and Supplementary Fig. 110) but also to the presence of robust
aluminum–oxygen bonds. Theoretical and experimental studies have
proved that the presence of high valent metals, high nuclearity, and
the presence of metal-oxygen bonds in metal clusters are key factors
influencing water stability[41–44].

The lattice void volumes for **AlMC-1** to **AlMC-7** are in a range of
16.9–49.0% using PLATON calculations. The $N_2$ sorption isotherms at
77 K were also obtained for **AlMC-1** to **AlMC-7**, and the calculated
Brunauer–Emmett–Teller (BET)-specific surface areas are, respec-
tively, 233.36, 132.87, 217.17, 182.39, 557.91, 370.47 and 161.04 $m^2 g^{-1}$
(Fig. 4e and Supplementary Fig. 111). The typical type-I isotherms for
them indicate the existence of micropores in the crystals (Supple-
mentary Fig. 112), suggesting they can be employed as potential
adsorbents. In addition, they remain stable after the adsorption tests
(Supplementary Fig. 113).

## Iodine absorption

**AlMC-1** was chosen as the ideal candidate for adsorption experiments
since it can be prepared on a large scale (Supplementary Figs. 114 and
115) and forms well-defined rectangular crystals (Fig. 5a) of uniform
size (~50 μm, passed through a 200-mesh sieve, 20 mg). An aqueous
solution of $I_2$/KI was chosen as the reaction medium because it can
function as an effective source of $I^-$, $I_2$ and $I_3^-$ based on the dynamic
equilibrium $I_2 + I^- \rightleftharpoons I_3^{-}$[13,45]. Colorless crystals of **AlMC-1** undergo a
noticeable color change within 1 minute by eye when immersed in this
solution (Fig. 5b, c, Supplementary Movie 3), turning black after
30 min. Single-crystal X-ray diffraction proves that the $NO_3^-$ and EtOH

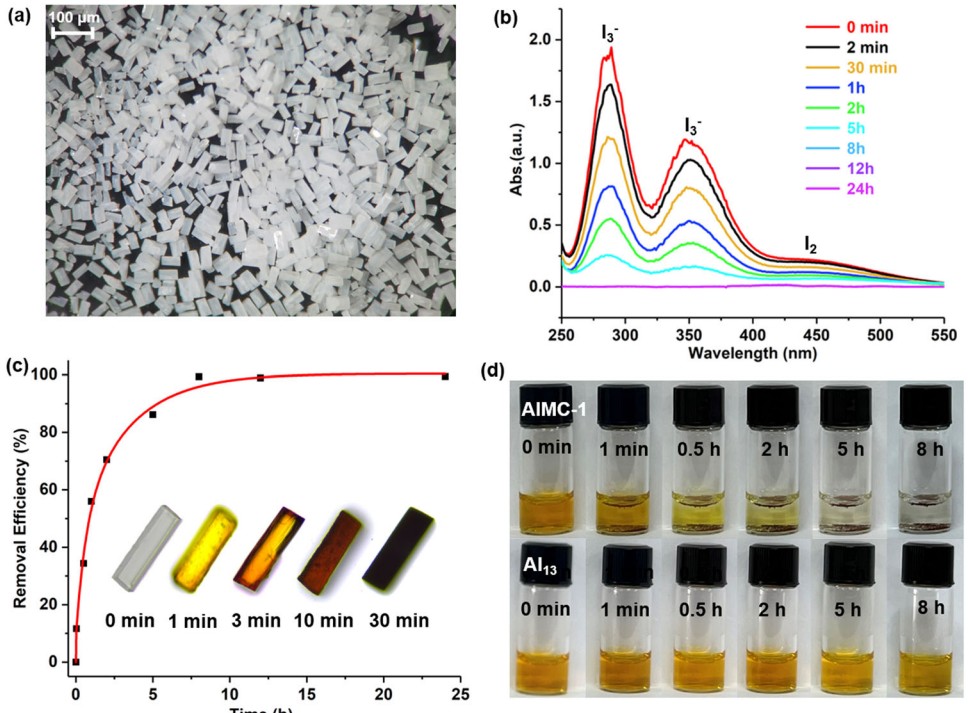

**Fig. 5 | Iodine removal by AlMC-1 crystals in 400 ppm I₂/KI aqueous. a** Uniformly sized single-crystal adsorbents (~50 μm) ready for removing iodine from water. **b** Time-dependent UV–Vis spectra upon the addition of **AlMC-1**. **c** The iodine adsorption removal efficiency is based on the absorption peak at 286 nm. Inset: The color change of single crystals during iodine adsorption (1 mL, 400 ppm). **d** the comparison of iodine adsorption between **AlMC-1** and the Al₁₃ cluster reported in the literature[23]. Source data are provided as a Source Data file.

guests in **AlMC-1** have been replaced by iodide ions after 30 min (Supplementary Fig. 116). The characteristic absorptions of $I_3^-$ and $I_2$ in the UV–vis spectra[46] in the aqueous solution of I₂/KI decrease in intensity with time until equilibrium is reached after 8 h, with the iodine removal efficiency being up to 99% (Fig. 5c). Using the cationic Al₁₃ cluster ([AlO₄Al₁₂(μ₂-OH)₁₂(OCH₂CH₂OH)₁₂]⁷⁺) with the same weight for comparison[23], **AlMC-1** exhibits a faster adsorption rate and a higher removal efficiency (Fig. 5d).

To explore the adsorption range further, we evaluated the iodine adsorption behavior of **AlMC-1** by using I₂/KI solutions with different concentrations (4, 40, 400, 2000 to 100,000 ppm). In all cases, the color of the crystals exhibited a noticeable change after 48 h adsorption, and the color is deeper at this point with higher concentration (Supplementary Figs. 117–123). The iodine-loaded crystalline samples were also characterized by EDS, X-ray photoelectron spectroscopy (XPS), Raman, and PXRD. The EDS and XPS results show increases in the amount of iodine absorbed with increased concentration of the I₂/KI solutions used (Supplementary Figs. 124 and 125). As can be seen from the expanded XPS spectra (Fig. 6a), the two peaks for the I3$d$ transition move to the higher binding energy with increased iodine loading, indicating an enhanced degree of aggregation[47,48]. The intensities of the Raman signals also increase with the concentration of the I₂/KI solution used (Fig. 6b). The band of 110 cm⁻¹ can be attributed to the symmetric stretching vibration of $I_3^-$ [49], the peak of 150 cm⁻¹ belongs to the asymmetric stretching vibration of $I_3^-$, and the peak at 220 cm⁻¹ is attributed to the I–I stretching vibration. The latter is different from "free" I₂ dissolved in a nonpolar solvent (~211 cm⁻¹)[50], indicating that the confined I₂ molecules in **AlMC-1** have strong interactions with the host. In addition, with increased iodine loading, new diffraction peaks in the PXRD can be observed, indicating that there is a significant host-guest interaction between iodine species and the host lattice (Fig. 6c)[15].

In order to understand the capture process on a microscopic level, we carried out mechanistic studies using single-crystal X-ray diffraction on crystals obtained from the above iodine absorption experiments at different iodine concentrations. The results from this detailed analysis provide 'snapshots' of the entry of iodine species into the host lattice and Al₂₄ unit as the concentration is increased and are shown in Fig. 6d. Porous **AlMC-1** provides several sites for iodine incorporation, with the adsorption sites in the crystal lattice being gradually occupied with increased concentration from intermolecular to intramolecular. The iodine guests first enter the hydrophobic intermolecular channels (site 1, 4–400 ppm, C-H···I interactions: 2.317–4.067 Å and I···π interactions: 3.279–4.575 Å) and the intermolecular μ₂-OH windows (site 2, 4 ppm, O-H···I: 2.887–3.196 Å) (Supplementary Figs. 126–129), then into the Al₈ macrocyclic units of the Al₂₄ units (site 3, 2000 ppm, C-H···I interactions: 2.820–3.387 Å and I···π interactions: 4.799–4.885 Å) (Supplementary Fig. 130), and finally occupying the interior cavity of the Al₂₄ units (site 4, 100,000 ppm, O-H···I interactions: 2.928–3.113 Å) (Supplementary Fig. 131). The exact nature of all of the iodine species present in the lattice at every stage cannot be deduced unambiguously from the X-ray data due to the disordering of the iodine sites. At 4 ppm I₂/KI$_{(aq)}$, the intermolecular channels can capture $I_3^-$ ions (Supplementary Fig. 126b), and the occupancy of these increases with concentration (40 ppm) (Supplementary Fig. 127a). Then iodine atoms continue to accumulate at site 1 until they reach saturation (Supplementary Fig. 128). Subsequently, I₂ molecules and disordered iodine species appear at site 3 (2000 ppm) (Supplementary Fig. 130e) and site 4 (100,000 ppm) (Supplementary Fig. 131e). It can be noted that the simulated PXRD patterns are consistent with experimental patterns for the iodine-loaded samples, indicating the evolution of the iodine species proposed analysis from SCXRD is rational (Supplementary Figs. 132–136).

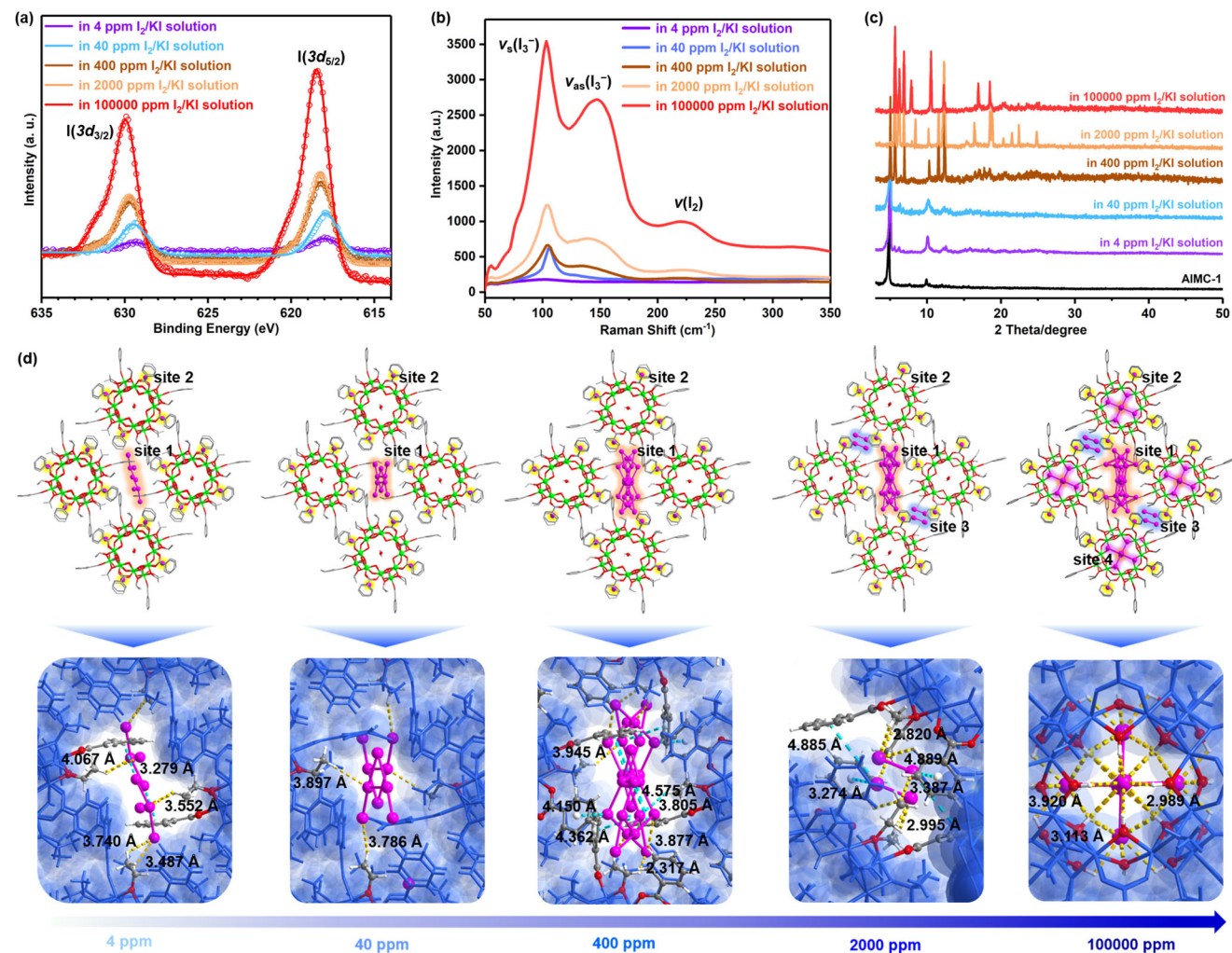

**Fig. 6 | Characterization and mechanism of the iodine capture processes of AlMC-1 in 4–100,000 ppm I$_2$/KI aqueous solutions.** Macroscopic characterization of **AlMC-1** after immersing in 4–100,000 ppm I$_2$/KI aqueous solutions, including **a** expanded XPS spectra, **b** Raman spectra, and **c** PXRD patterns. **d** Mechanistic studies at the molecular level (top row: adsorption site; bottom row: corresponding supramolecular interactions between iodine species and Al$_{24}$ cage hosts; site 1 intermolecular channel; site 2 square window of O$_{24}$-***tro*** cage; site 3 Al$_8$ macrocyclic cavity; site 4 the O$_8$ cavity of Al$_{24}$ cages). The strong hydrogen-bond interactions (C−H···I and O−H···I) are represented by yellow dotted bonds, while the I···π interactions are expressed by blue dotted bonds (Al: bright green; O: red; C: gray; H: white; N: blue; Cl: sea green; Br: purple; I: pink). Source data are provided as a Source Data file.

Such a low-concentration iodine/iodide capture ability suggests that **AlMC** compounds of the type described in this paper may be of value in the capture of environmental iodide/iodine (e.g., waste-water streams). We evaluated the adsorption capacity of **AlMC-1** using a 100,000 ppm I$_2$/KI aqueous solution based on the results of single-crystal analysis, which is about 820.3 mg/g. This value is slightly lower than that obtained from gravimetric (1.03 g/g) and titrimetric analysis (890 mg I$_2$/g)[13,15,51], which may be due to the disorder of the iodine species in the lattice and difficulty in determining the precise composition by X-ray single-crystal analysis. Even so, the capacity of **AlMC-1** is still higher than that of MOFs (Supplementary Table 9)[52,53] and noble metal-based adsorbents[54–56]. The iodine-loaded crystals of **AlMC-1** can be reused as iodine sponges after the removal of the iodine components by washing them with HOEt (Supplementary Fig. 137). PXRDs of **AlMC-1** after 200 kGy β or γ irradiation also indicate no structural degradation (Supplementary Fig. 138a), and the retention of the adsorption capacity compared to non-irradiated samples (Supplementary Fig. 138b). This is important in regard to the potential applications in the removal of radioactive iodine.

In this paper, we explored the structural landscape of a series of unique solid-state materials based on a cationic Al$_{24}$ Archimedean host. The highly symmetrical Al$_{24}$ cage has a purely inorganic ***tcu*** kernel and six calixarene-like Al$_8$ shells. The core−shell arrangement is highly adaptive toward a variety of guests (NO$_3^-$, OEt$^-$, O$^n$Pr$^-$, Cl$^-$, Br$^-$, and I$^-$), generating a broad range of supramolecular lattice arrangements in the solid-state. The unique structural features (simultaneously containing hydrophobic outer channels and a hydrophilic inner cavity) make these materials highly stable in water. Iodine/iodide capture experiments have revealed rapid enrichment, low-concentration capture, high adsorption capacity, recyclability, and radiation-resistant characteristics for **AlMC-1**, indicating its potential applications in trace iodine extraction in waste-water streams (such as in the radiation industry). These host assemblies provide alternatives to organic hosts such as calixarenes, crown ethers, and pillararenes, for water purification.

## Methods

### Syntheses of AlMC compounds

**AlMC-1** to **AlMC-7** were synthesized by mixed Al(O$^i$Pr)$_3$ (6 mmol), benzoic acid (3 mmol), HNO$_3$ (60–100 μL) in 8–10 mL alcohols solvent (HOEt or HO$^n$Pr). The trace of HNO$_3$ plays an important role in the formation of the Al$_{24}$ cages. Its use should be controlled between 60

and 100 μL in this reaction. Besides, the addition of extra guests greatly affects the supramolecular assembly of $Al_{24}$ cages, such as $H_2O$, benzyl alcohol, quaternary ammonium salt and pyrazole. Notably, water helps to increase the yield, for example, it increased from ~11% to ~43% when 60 μL $H_2O$ is introduced in the reaction system of **AlMC-1**. See Supplementary Methods for more details on the synthesis of all of the compounds described in this paper.

## Scale-up synthesis of AlMC-1

A mixture of $Al(O^iPr)_3$ (6.0 g), benzoic acid (1.8 g), $HNO_3$ (0.5 mL), $H_2O$ (300 μL), and ethanol (40 mL) was sealed in an 80 mL vial and heat at 80 °C for 7 days. When cooled to room temperature, the white precipitate and colorless crystals are washed by ethanol repeatedly. After drying, the precipitate and crystals are passed through a 200-mesh sieve, and pure-phase rectangular crystals are obtained. (Yield: ~817 mg, ~16% based on $Al(O^iPr)_3$).

## Iodine adsorption

Crystals with moderate size were selected for iodine adsorption research. Large crystals tend to fracture during adsorption, while the X-ray diffraction intensity for tiny-size crystals is very weak. Thus, we choose ~15 μm × 50 μm crystals for iodine adsorption measurements. A 20 mg sample was immersed in $I_2$/KI aqueous (10 mL) with various concentrations for 48 h at room temperature. The iodine-loaded samples obtained were filtered and washed prior to characterizations (XPS, EDS, Raman, and PXRD) and SCXRD. To measure the adsorption capacity, **AlMC-1** crystalline samples (50 mg) were soaked in 100,000 ppm $I_2$/KI$_{(aq)}$ (300 mg KI and 300 mg $I_2$ in 3 mL $H_2O$) for 48 h. The iodine-saturated samples were collected by filtration, washing with water (2 mL × 30 times) until the filtrate became clear, and dried in air for gravimetric analysis and characterization. The filtrate was collected and 2 mL 2% aqueous starch indicator was added for sodium bisulfite titration analysis.

## Regeneration and recycling experiment

Iodine-loaded samples were immersed in HOEt (2 mL) for desorption of the iodine/iodide, during which the solvent was decanted and washing repeatedly several times. The desorption was observed to take place rapidly. Once HOEt is added, the color of the solution turned yellow immediately and then gradually deepened. This process can be investigated using time-dependent UV–Vis spectra.

## Irradiation stability measurements

**AlMC-1** (100 mg) was irradiated at a dose rate of 20 kGy/h for 10 h using a $^{60}$Co β- or γ-irradiation source. β-Irradiation was provided by an electron accelerator located at the CGD Dasheng Electron Accelerator Co., Ltd., in Jiangsu Province, China. While γ-irradiation was conducted by Gansu Tianchen Irradiation Technology Co., Ltd., in Gansu Province, China.

**Caution.** $HNO_3$ is corrosive and has a pungent odor. Thus, the experiments should be carried out in a fume hood, and gloves and masks should be worn.

## Data availability

X-ray crystallographic data for the structures reported in the article have been deposited at the Cambridge Crystallographic Data Centre, under deposition numbers CCDC 2193096 (**AlOC-60**), 2193097 (**AlMC-1**), 2193098 (**AlMC-2**), 2193099 (**AlMC-3**), 2193100 (**AlMC-4**), 2193101 (**AlMC-5**), 2193102 (**AlMC-6**), 2193103 (**AlMC-7**), 2193104 (**AlMC-8**), 2193105 (**AlMC-1a**), 2193106 (**I@Al$_{24}$−400 ppm−30 min**), 2193107 (**I@Al$_{24}$−4 ppm−48 h**), 2193108 (**I@Al$_{24}$−40 ppm−48 h**), 2193109 (**I@Al$_{24}$−400 ppm−48 h**), 2193110 (**I@Al$_{24}$−2000 ppm−48 h**) and 2193111 (**I@Al$_{24}$−100,000 ppm−48 h**). Copies of the data can be obtained free of charge via https://www.ccdc.cam.ac.uk/ structures/.

The dataset is also provided as Supplementary Data 1 with this paper. All other data supporting the findings of this study are available within the paper, its supplementary information, or the corresponding author. Source data are provided with this paper.

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

## Acknowledgements

This work is supported by the National Natural Science Foundation of China (92061104, 21935010), Natural Science Foundation of Fujian Province (2021J06035), and Youth Innovation Promotion Association CAS (2017345 and Y2021081).

## Author contributions

W.-H.F. and J.Z. conceived and designed this project. Y.-J.L. carried out the synthesis, characterization, and iodine adsorption study. Y.-F.S. assisted with sample characterization and data analysis. S.-H.S. and Z.-H.L. gave support on the scale-up synthesis. S.-T.W. assisted with structural determination. W.-H.F., J.Z., D.S.W., and Y.-J.L. wrote the manuscript. All the authors discussed the results and commented on the manuscript.

## Competing interests

The authors declare no competing interests.

## Additional information

**Supplementary information** The online version contains
supplementary material available at

Wei-Hui Fang or Jian Zhang.

**Peer review information** *Nature Communications* thanks Xiaodong Chi,
Baiyan Li and the other, anonymous, reviewer(s) for their contribution to
the peer review of this work. Peer reviewer reports are available.

