## [Peer Review File · Nature Communications]

Water-stable Porous Al₂₄ Archimedean Solids for Removal of Trace IodineREVIEWER COMMENTS

Reviewer #1 (Remarks to the Author):

This manuscript describes a new type of core-shell crystalline material based on inorganic Archimedean truncated hexahedron (tcu) polyhedron zeolitic cage structure. Different guests can cause different lattice pore shapes in the crystalline state for its unique structures. More importantly, this material can remove trace levels of iodine from water effectively. The structure of this material and the crystals with different iodine are interesting. This manuscript is well written and clearly presented the innate feature of the advantages of polyhedron zeolitic cage for the iodine capture. Thus, I highly recommend its publication in Nature Communications after minor revision.

1. In Fig. 1c, the ion above and below the arrow needs to turn over.
2. The author said that "The presence of some of the anion guests can be confirmed by energy-dispersive X-ray dispersive spectroscopy (EDS)", which surface of the crystal was used to carry out the EDS mapping and whether different surfaces will have different element content? Why not use elemental analysis to identify the presence of some of the anion guests?
3. The thermogravimetric analysis can give more information in thermal stability.
4. The author mentioned that all the compounds are stable in different solvents, however, in Fig. S87, the experiment only proved that AIMC-1 is stable in different solvents, what about AIMC-2 to AIMC-7? Moreover, the ESI-MS just demonstrate the stability in CH₃CN, does the compounds still stable in DMF or DMSO? ¹H NMR and ESI-MS in DMF or DMSO should be tested. As all the compounds can dissolve in CH₃CN, why the ESI-MS of AIMC-7 was tested in the mixture of CH₃CN and DMF? And what's the ratio of CH₃CN and DMF?
5. The N₂ sorption isotherms at 77 K are also need to be obtained for AIMC-2, AIMC-3 and AIMC-7. What's the Fig. S96 used for? Please explain Fig. S96 in another sentence.
6. In Fig. 5b, the decrease of UV-vis absorbance value between 30 mins and 1 hour is more than the decrease value between 2 mins and 30 mins, what caused this result?
7. In Fig. S112, the caption of (c) and (d) was lost. Besides, after adsorption for 48 h, more iodine placed on the site 1 when compared with that of 30 mins adsorption, does this mean that the iodine captures in site 2 is faster than site 1 rather than simultaneously? In addition, please update the crystal structure of immersing AIMC-1 in 400 ppm aqueous I₂/KI after 30 mins.
8. A bipyridine based cage has also been developed for iodine capture in vapor and organic solvent mediums, as well as aqueous phase (J. Am. Chem. Soc. 2022, 144, 113–117). This should be cited in the introduction part of supramolecular materials and cage compounds for iodine capture so that the readers could have a deeper understanding of the background.

Reviewer #2 (Remarks to the Author):

In this manuscript, the authors report the synthesis of Al₂₄ cage and its application for iodine adsorption from water. The structure of the Al₂₄ cage and their supramolecular assemblies generated under different synthetic conditions were characterized and investigated. Among these supramolecular assemblies, AIMC-1 exhibits good stability towards water and moderate porosity with BET surface area of 233.36 m² g⁻¹. AIMC-1 can capture iodine from concentrated aqueous I₂/KI solution with an adsorption capacity of 1.03g/g and be regenerated after washing with ethanol. However, the capacity of iodine capture is not attractive enough and is lower compared to other porous materials. Thus, the work does not deserve a publication in high level journal like Nat. Commun.

1. In Figure S96, S120, and S121, it is better to compare the PXRD patterns of AIMC-1, AMIC-5, AIMC-6 after adsorption/desorption/irradiation with the ones before adsorption/desorption/irradiation, instead of comparing with the simulated patterns.
2. In Figure S120 (d), the adsorption capacity of AIMC-1 drops significantly after three cycles, I wonder what is the iodine release efficiency of I@Al₂₄-6
3. There are several grammar mistakes existing in this manuscript. Please read carefully and correct all of them.

Dear reviewers,

We sincerely appreciate your timely review and constructive comments which helped us to further improve the manuscript. According to these comments and suggestions, we have carefully modified the original manuscript. All the changes in the manuscript are marked in yellow. Here are our point-by-point responses:

Response to the reviewer's comments

REVIEWER COMMENTS

Reviewer #1 (Remarks to the Author):

This manuscript describes a new type of core-shell crystalline material based on inorganic Archimedean truncated hexahedron (tcu) polyhedron zeolitic cage structure. Different guests can cause different lattice pore shapes in the crystalline state for its unique structures. More importantly, this material can remove trace levels of iodine from water effectively. The structure of this material and the crystals with different iodine are interesting. This manuscript is well written and clearly presented the innate feature of the advantages of polyhedron zeolitic cage for the iodine capture. Thus, I highly recommend its publication in Nature Communications after minor revision.

1. In Fig.1c, the ion above and below the arrow needs to turn over.

Response: Thank you very much for your careful review, this point has been corrected in the revised manuscript.

2. The author said that “The presence of some of the anion guests can be confirmed by energy-dispersive X-ray dispersive spectroscopy (EDS)”, which surface of the crystal was used to carry out the EDS mapping and whether different surfaces will have different element content? Why not use elemental analysis to identify the presence of some of the anion guests?

Response: Yes, different surfaces have been selected for analysis to verify the existence of all elements. According to your suggestion, the elemental analysis for AIMC-1 to AIMC-7 has been provided (Page S5 and S6).

3. The thermogravimetric analysis can give more information in thermal stability.

Response: Yes, we agree with you. The thermogravimetric analyses (TGA) for AIMC-1 to AIMC-7 have been provided (Figure S86–S92, Page S43–S45), indicating that all of them can be kept

stable up to 170–180 °C. And the cage will be decomposed after 180 °C.

4. The author mentioned that all the compounds are stable in different solvents, however, in Fig. S87, the experiment only proved that AIMC-1 is stable in different solvents, what about AIMC-2 to AIMC-7? Moreover, the ESI-MS just demonstrate the stability in CH₃CN, does the compounds still stable in DMF or DMSO? ¹H NMR and ESI-MS in DMF or DMSO should be tested. As all the compounds can dissolve in CH₃CN, why the ESI-MS of AIMC-7 was tested in the mixture of CH₃CN and DMF? And what's the ratio of CH₃CN and DMF?

Response: According to your comments, the stability for AIMC-2 to AIMC-7 in different solvent have been provided (Figure S95–S100, Page S47–S50). Although all of them can keep stable in water, they behave differently in organic solvents. **AIMC-1** and **AIMC-2** can keep stable in organic solvents with medium or low polarity, such as cyclohexane (polarity: 0.1), CH₂Cl₂ (polarity: 3.4), ethanol (polarity: 4.3), ethyl acetate (polarity: 4.3), 1,4-dioxane (polarity: 4.8). While, **AIMC-3** to **AIMC-7** will be dissolved in CH₂Cl₂, ethanol etc. solvents with medium polarity, and can only keep stable in petroleum ether (polarity: 0.01), n-hexane (polarity: 0.06), cyclohexane (polarity: 0.1), methylbenzene (polarity: 2.4).

Considering the low boiling point of CH₃CN compared with DMF and DMSO, hence we provided the ESI-MS in CH₃CN. The Al₂₄ cage can also keep stable in DMF and DMSO. According to your suggestion, we have also measured the ESI-MS and ¹H NMR in DMSO or DMSO-d₆ (Figure S101, Page S50 and S51; Figure S105, Page S67). Both of them indicate the good solubility stability of Al₂₄ cage. The solubility of **AIMC-7** in CH₃CN is not very good, thus, we add a drop of DMF to help dissolve. The ratio of CH₃CN and DMF is 20:1, which has been added in the revised manuscript (Page 8, legend).

5. The N₂ sorption isotherms at 77 K are also need to be obtained for AIMC-2, AIMC-3 and AIMC-7. What's the Fig. S96 used for? Please explain Fig. S96 in another sentence.

Response: The N₂ sorption isotherms at 77 K for **AIMC-2**, **AIMC-3** and **AIMC-7** have been provided (Figure S111, Page S69). Figure S113 (Original Figure S96) is used to indicate that the crystals remain stable after the N₂ adsorption tests. And the related description has been added in the manuscript (Page 7).

6. In Fig. 5b, the decrease of UV-vis absorbance value between 30 mins and 1hour is more than the decrease value between 2 mins and 30 mins, what caused this result?

Response: This may be caused by the contamination of the cuvette. We have retested the time-dependent UV-Vis spectra and found a significant difference when adsorption 2 min. We have updated this curve in Figure 5b and the related removal efficiency in Figure 5c has been also corrected in the revised manuscript (Page 9).

7. In Fig. S112, the caption of (c) and (d) was lost. Besides, after adsorption for 48 h, more iodine placed on the site 1 when compared with that of 30 mins adsorption, does this mean that the iodine captures in site 2 is faster than site 1 rather than simultaneously? In addition, please update the crystal structure of immersing AIMC-1 in 400 ppm aqueous I₂/KI after 30 mins.

Response: The caption of (c) and (d) has been added in Figure S129 (Original Figure S112, Page S79). Indeed, a pair of H₂O guests on μ_2 -OH square window is replaced by I⁻ when adsorption 48h rather than 30 min. Combined with the adsorption data at different concentrations, these two H₂O guests are indeed hard to be substituted by iodine ion at low concentration I₂/KI solution or in short adsorption time, which may be owing to their more closed cavity when compared with EtOH and NO₃⁻ (Figure S27, Page S19) and strong hydrogen bonds in μ_2 -OH square window. Therefore, we agree with you that the iodine capture in the intermolecular channel is faster than that in μ_2 -OH square windows. The description (Page S78 and S79) and crystal structures have been updated in the manuscript and supplementary information.

8. A bipyridine based cage has also been developed for iodine capture in vapor and organic solvent mediums, as well as aqueous phase (J. Am. Chem. Soc. 2022, 144, 113–117). This should be cited in the introduction part of supramolecular materials and cage compounds for iodine capture so that the readers could have a deeper understanding of the background.

Response: This reference has been cited in the revised manuscript (Ref. 20 and Page 12).

Reviewer #2 (Remarks to the Author):

In this manuscript, the authors report the synthesis of Al₂₄ cage and its application for iodine adsorption from water. The structure of the Al₂₄ cage and their supramolecular assemblies generated under different synthetic conditions were characterized and investigated. Among these supramolecular assemblies, AIMC-1 exhibits good stability towards water and moderate porosity with BET surface area of 233.36 m² g⁻¹. AIMC-1 can capture iodine from concentrated aqueous I₂/KI solution with an adsorption capacity of 1.03g/g and be regenerated after washing with ethanol.

However, the capacity of iodine capture is not attractive enough and is lower compared to other porous materials. Thus, the work does not deserve a publication in high level journal like Nat. Commun.

Response: We have to admit the iodine capture capacity for **AIMC-1** is not that high compared with other types of materials. However, it cannot be used as the only criterion for whether an article deserves publication in a high-level journal. There are lots of excellent works with low adsorption capability that have been published and received wide attention. For example, HILS zeolite with a capability of 0.53 g/g (*Energy Environ. Sci.* 2016, 9, 1050), EtP6 with a capacity of 0.25g/g (*J. Am. Chem. Soc.* 2017, 139, 15320), {[Zn₃(DL-lac)₂(pybz)₂]}·2.5DMF with the capability of 1.01 g/g (*J. Am. Chem. Soc.* 2010, 132, 2561), molybdenum sulfide porous chalcogel with a adsorption capacity of 1 g/g (*J. Am. Chem. Soc.* 2015, 137, 13943). On the contrary, there are also a lot of innovative works that concentrate on synthesis, structure, and conceptual innovations, and are published in high-level journals (*Science* 2011, 333, 436; *Nature* 2019, 565, 213; *Nat. Chem.* 2014, 6, 906; *Nat. Chem.* 2014, 6, 673).

Our work is innovative from three perspectives including synthesis, structure, and concept. Firstly, in terms of synthesis, we realized the structural transformation from a neutral aluminum-oxo molecule ring to a cationic aluminum-oxo porous cage by adding nitric acid for the first time. Secondly, the porous inorganic Archimedes *tcu* kernel and organic calixarene-like cavities combine the characteristics of molecular sieves and macrocycles. Lastly, we proposed a new concept for these special materials as “aluminum macrocycle-faced cages (AIMC)”. In addition, these materials have good water stability, can be prepared in large quantities, and enable rapid iodine capture from low concentration aqueous I₂/KI solutions (down to 4 ppm concentration). Besides, this work also revealed the detailed iodide ion capture process and specific interactions through single crystal X-ray diffraction, which is very valuable and will provide more ideas for the design and synthesis of high-performance iodine-adsorbent materials. In conclusion, our work deserves to be published in Nat. Commun.

1. In Figure S96, S120, and S121, it is better to compare the PXRD patterns of AIMC-1, AMIC-5, AIMC-6 after adsorption/desorption/irradiation with the ones before adsorption/desorption/irradiation, instead of comparing with the simulated patterns.

Response: The PXRD patterns before adsorption/desorption/irradiation have been added in Figure

S113 (Page S71), S137 (Page S85) and S138 (Page S85). The PXRD patterns after adsorption/desorption/irradiation all matched well with the original crystals, indicating that they are stable during these tests.

2. In Figure S120 (d), the adsorption capacity of AIMC-1 drops significantly after three cycles, I wonder what is the iodine release efficiency of I@Al₂₄-6.

Response: According to your suggestion, we have calculated the release efficiency. The adsorption capacity of AIMC-1 drops significantly after three cycles resulting from the calculated release efficiency is about 84.8% (Page S85).

3. There are several grammar mistakes existing in this manuscript. Please read carefully and correct all of them.

Response: We have carefully examined and revised the grammar mistakes throughout the manuscript.

Reviewer #3 (Remarks to the Author):

This is the most interesting molecular structure I've seen this year. In this paper, Liu et al. report an interesting evolution from aluminum molecular ring to Al₂₄ truncated hexahedron cage. Using its calixarene-like structure, the cage can capture different types of guests, including alkoxide, halogen ions, nitrate, and small clusters. The authors analyze the host-guest interactions in eight prepared Al₂₄ cages in detail and show their fascinating packing modes. All of the figures in the manuscript are vivid and easy to understand. Besides, the sufficient stability characterizations and scale-up synthesis make it convincing that this cage is expected to play a role in some practical applications. Of course, the authors evaluate the iodine adsorption capability of the cage in water and reveal the adsorption mechanism through single-crystal X-ray diffraction. Obviously, the adsorption sites are different from those MOF materials and traditional precious metal materials, thus, that may provide new ideas for the design of efficient adsorption materials. The figures in the manuscript and supplementary information are generally well laid out. The subject and novelty are appropriate for Nat. Commun., and it will be interesting to those interested in the structural chemistry of cage materials and framework materials. I am therefore happy to suggest this manuscript is suitable for publication in Nat. Commun., and only provided some minor comments for modification:

1 Although the authors provided the reaction and crystal photos of scale-up synthesis, the purity of

the samples still needs to be considered. Generally, scale-up synthesis is not easy. So, please provide the PXRD patterns of AIMC-1 obtained from larger-scale synthesis to assure purity.

Response: Thank you for your comments. We have added the PXRD pattern of AIMC-1 obtained from larger-scale synthesis (Figure S115, Page S72). It matched well with simulated patterns, indicating the phase purity of the sample.

2 Have authors tried to use larger monocarboxylic acid ligands to construct Al₂₄ cage. Such as 4-biphenylcarboxylic acid or p-terphenyl-4-carboxylic acid. These ligands can make the calixarene-like cup larger and may be conducive to capturing larger guests.

Response: Thank you for your suggestion. It is a good idea to enlarge the calixarene-like cup-like reticular chemistry in MOF and facilitate capturing larger guests. And we have tried these ligands before submission and during the revision period. However, we have not obtained any crystalline product, which may be owing to the poor solubility of these ligands. We do believe this idea will realize in the future.

3 For Figure S86, it is better to combine thermogravimetry to determine the exact temperature range in which the cage is stable. Besides, it is important to confirm if there is any significant weight loss before 150 °C due to the volatile characteristics of alkoxides.

Response: Thank you for your comments. The thermogravimetry analysis for AIMC-1 to AIMC-7 has been provided in Figure S86–S92 (Page S43–S45). AIMC-1 can keep stable to about 170 °C, after that a large proportion of weight loss will occur. Indeed, as you say, there is 5.17% weight loss before 170 °C that can be assigned to the release of guests and unresolved ethanol molecules.

Sincerely

Yours

Prof. Wei-Hui Fang

Fujian Institute of Research on the Structure of Matter Chinese Academy of Sciences,
Fuzhou, Fujian 350002, P. R. China.

E-mail: fwh@fjirsm.ac.cn

REVIEWER COMMENTS

Reviewer #1 (Remarks to the Author):

This revised manuscript had carefully modified the original manuscript according to the comments and suggestions raised by reviewers. It clearly showed the innovative from three perspectives including synthesis, structure, and concept. Moreover, this work also revealed the detailed iodide ion capture process and specific interactions through single crystal X-ray diffraction, which is very valuable and will provide more ideas for the design and synthesis of high-performance iodine-adsorbent materials. Thus, I highly recommend its publication in Nature Communications without further revision.

Reviewer #3 (Remarks to the Author):

I recommend this paper to be accepted.